# BEYOND SYNTAX:
# ACTION SEMANTICS LEARNING FOR APP AGENTS

## ABSTRACT

The recent development of Large Language Models (LLMs) enables the rise of App agents that interpret user intent and operate smartphone Apps through actions such as clicking and scrolling. While prompt-based solutions with proprietary LLM APIs show promising ability, they incur heavy compute costs and external API dependency. Fine-tuning smaller open-source LLMs solves these limitations. However, current supervised fine-tuning methods use a syntax learning paradigm that forces agents to reproduce exactly the ground truth action strings, leading to out-of-distribution (OOD) vulnerability. To fill this gap, we propose **Action Semantics Learning (ASL)**, a novel learning framework, where the learning objective is capturing the semantics of the ground truth actions. Specifically, inspired by the programming language theory, we define the action semantics for App agents as the state transition induced by the action in the user interface. Building on this insight, ASL employs a novel **SEmantic Estimator (SEE)** to compute a semantic similarity to train the App agents in generating actions aligned with the semantics of ground truth actions, even when their syntactic forms differ. SEE is a flexible module that can be applied in both supervised and reinforcement fine-tuning paradigms. To support the effectiveness of ASL, we theoretically demonstrate the superior robustness of ASL for the OOD problem compared with the existing syntax learning paradigm. Extensive experiments across multiple offline and online benchmarks demonstrate that ASL significantly improves the accuracy and generalisation of App agents compared to existing methods.

## 1 INTRODUCTION

The rapid development of Large Language Models (LLMs) [1] has led to the emergence of AI agents across a wide range of domains (Putta et al., 2024) Li et al. (2023); Gou et al. (2023). Among these, AI agents for smartphone App operation, commonly referred to as App agents, are gaining significant attention for leveraging LLMs to enable fully automated App navigation (Rawles et al., 2024b;a; Wang et al., 2024a; Chen et al., 2024). These agents interpret user-provided goals in natural language and execute tasks such as scheduling, messaging, and online purchasing by interacting with the user interfaces (UIs) via pre-defined actions, such as tapping, scrolling, or entering text. The diversity of smartphone operating systems and UI descriptions, coupled with predefined action sets for specific systems, poses a significant challenge in developing generalisable App agents.

Recent advances in App agents can be broadly grouped into two approaches. One prominent approach leverages proprietary LLMs APIs, such as GPT (Hurst et al., 2024) and Gemini (Hassabis & Kavukcuoglu, 2024), through *prompt engineering* techniques to comprehend user intents and device states, subsequently generating suitable actions (Rawles et al., 2024a; Li et al., 2024). While these App agents exhibit robust reasoning and generalisation capabilities, their practical deployment is often hindered by challenges such as high inference latency, substantial computational resource requirements, and frequent dependence on third-party APIs, which may escalate operational costs and introduce instability. Alternatively, *fine-tuning* smaller LLMs with offline datasets or interactive online trajectories has emerged as a viable strategy, achieving low-latency inference conducive to real-world applications (Wang et al., 2024b; Christianos et al., 2024; Bai et al., 2024). However, current

---

[1] In this work, LLMs refer to foundation models that process multiple input modalities (e.g., visual or multimodal) and generate textual outputs.

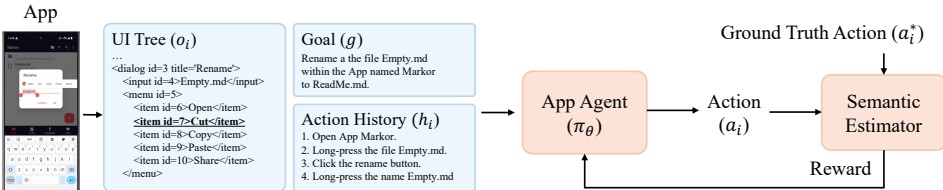

Figure 1: Our ASL framework. The reward corresponds to the function defined in Eq. (2) .

supervised methods typically follow a *syntax learning paradigm* that forces agents to reproduce exactly the ground truth action string for each UI within the training data, instead of understanding the underlying action semantics. This leads to an out-of-distribution (OOD) vulnerability: even the UIs slightly differ from the samples within the training data can provoke a dramatic performance drop. An agent trained to follow a specific interface path may fail when the UI changes or alternative options are presented. For example, an agent trained to delete text by generating "Tap the Delete button" will fail if the Delete button is unavailable, whilst the "Cut" button is still visible. See Sec. 3.1 for further discussion of this OOD vulnerability.

To go beyond syntax and capture action semantics, we first draw inspiration from the denotational semantics in programming language theory, which defines program semantics by its effect on system state (Stoy, 1981), to define the action semantics for App agents as the *state transition* induced by the action in the UI environment. Building on this insight, we propose **Action Semantics Learning (ASL)**, a novel learning framework that shifts the learning objective of App agents from exactly reproducing the ground truth action strings to reproducing their semantics. Specifically, during training, the actions generated by the App agent are rewarded by a **SEmantic Estimator (SEE)** that scores how closely the semantics of the generated action match that of the ground truth action. By employing the semantic reward, ASL considers actions that lead to the same UI state transition as the ground-truth action to be semantically equivalent, thereby guiding the agent to focus on semantics rather than syntactic form. In Sec. 3.4, we theoretically demonstrate that, compared with the syntax learning paradigm, ASL is more robust towards the OOD problem. See Fig. 1 for a visualisation of the proposed ASL framework.

The semantic estimator SEE is the core component of our learning framework. Specifically, SEE leverages an LLM-driven world model to predict the impact of an action on the current UI state and uses BERT to measure the semantic similarity between the effects of the predicted action and the ground truth action. Our design is inspired by recent world-model–empowered web agents (Chae et al., 2024), which leverage predicted observations as offline rewards for prompt engineering. In contrast, we are the first to incorporate world-model predictions directly into the app agent training stage, generating a semantic reward without requiring any extra computational resources for deployment.

Our main contributions are summarised as follows:

• We introduce ASL, a novel learning framework based on action semantics. In this framework, we formulate the App-agent training as the semantics-level learning: an action is correct if it produces the target UI state transition, regardless of its exact syntactic expression. This learning objective contributes to improving the generalisability of App agents.

• We propose SEE, a novel training-free semantic estimator that generates reward signals based on semantic alignment, enabling the fine-tuning of lightweight models to learn action semantics without additional GPU memory requirements for reward model training.

• We conduct extensive experiments on various online and offline benchmarks. Empirical results demonstrate that the proposed ASL framework achieves superior accuracy and generalisation compared to existing approaches, especially on more difficult tasks.

## 2 RELATED WORKS

**Prompt-driven App agents.** Prompt-driven App agents mark a substantial advancement in human–device interaction, enabling natural language commands to be translated into precise UI actions on smartphones, tablets, and other mobile devices. Recent research has progressively refined these

systems, with pioneering implementations like AppAgent (Zhang et al., 2023) demonstrating that large pretrained models can effectively map screenshots or accessibility trees to human-like actions. Building on these foundations, more recent frameworks have introduced advanced prompting strategies incorporating planning, self-reflection, and tool use to tackle increasingly complex tasks (Rawles et al., 2024a; Wang et al., 2024a). Despite their strong performance, current prompt-driven App agents remain heavily dependent on proprietary LLM APIs such as GPT and Gemini (Rawles et al., 2024a; Wang et al., 2024a; Zhang et al., 2023; Li et al., 2024). This dependence introduces several limitations, including high inference latency, substantial computational costs, and reliance on third-party services that raise concerns around data privacy, service reliability, and vendor lock-in. These challenges hinder real-world deployment and highlight the need for open, efficient, and controllable alternatives.

**Fine-tuned App agents.** Recent research has focused on fine-tuning smaller models tailored for mobile environments. LiMAC (Christianos et al., 2024) introduces a compact action transformer that predicts UI-grounded actions paired with a fine-tuned vision–language model for textual reasoning, while InfiGUIAgent (Liu et al., 2025b) adopts a two-stage approach that first learns UI semantics from screenshots and then generates actions conditioned on user instructions. Despite architectural differences, these supervised fine-tuning methods follow a syntax learning paradigm that forces agents to reproduce the exact ground truth action string. This paradigm can lead to OOD vulnerability: even slight UI differences from the training data samples can provoke dramatic performance degradation. Complementing these offline approaches, several works explore online optimisation to improve generalisability and robustness to OOD data. DigiRL (Bai et al., 2024) proposes a reinforcement learning framework that simulates App control scenarios and fine-tunes policies from an offline-supervised starting point. DistRL (Wang et al., 2024b) enhances this approach with asynchronous online training, boosting sample efficiency. However, such online methods typically assign rewards based on sparse binary outcomes from the environment, without assessing individual steps. In contrast, our SEE module introduces step-level evaluation, providing finer-grained and smoother feedback at each step rather than relying exclusively on environment-level rewards.

**World model boosted App agents.** World models simulate future states of the environment based on previous states and given actions (LeCun, 2022; Ding et al., 2024). In domains such as robotics, world models (Yang et al., 2023; Zhen et al., 2024; Zhou et al., 2024b; Ajay et al., 2023) can predict future visual states, such as images or videos across diverse environments. These predictive capabilities are critical for enabling robots to understand their surroundings, make informed decisions, and accurately execute tasks. Building on these advances, researchers have extended world modelling to web environments, where models (Chae et al., 2024; Gu et al., 2024; Liu et al., 2023) generate textual predictions of a website's future state following a user interaction, enabling the development of more accurate web agents. However, existing applications of world models in this setting are typically utilised for providing feedback at test time, which introduces additional inference overhead and remains suboptimal for practical deployment. In our work, we instead leverage world models as a reward signal to fine-tune the agent by computing the semantic similarity between predicted and ground truth actions, requiring no additional inference time or computational resources in deployment.

## 3 METHODOLOGY

In this section, we begin with the problem formulation, which introduces the smartphone app operation task and summarises the key challenge for current fine-tuning methods. Then, we propose a novel learning framework named Action Semantic Learning (ASL) for App agents. Furthermore, we elaborate on the semantic estimator (SEE), the key component within the ASL framework. Finally, we conduct a theoretical analysis to support the effectiveness of the ASL framework.

### 3.1 PROBLEM FORMULATION

**Smartphone operation**. We formulate smartphone operation as a Partially Observable Markov Decision Process defined by $(\mathcal{S}, \mathcal{A}, \mathcal{T}, \mathcal{R}, \mathcal{O}, \gamma)$, where $\mathcal{S}$ represents smartphone system states, $\mathcal{A}$ denotes actions for operating UIs, $\mathcal{T}$ is the state transition function, $\mathcal{R}$ is the utility function, $\mathcal{O}$ maps states to UI observations, and $\gamma$ is the discount factor. At each timestep $t$, the App agent, which is parameterised by $\theta$ and denoted as $\pi_\theta$, receives an observation $o_t$ of the current UI state, the user goal $g$ in natural language, and action history $h_t$. The App agent produces an action according to $a_t^\theta = \pi_\theta(g, h_t, o_t)$, where the observation $o_t$ may include a subset of available screenshots and

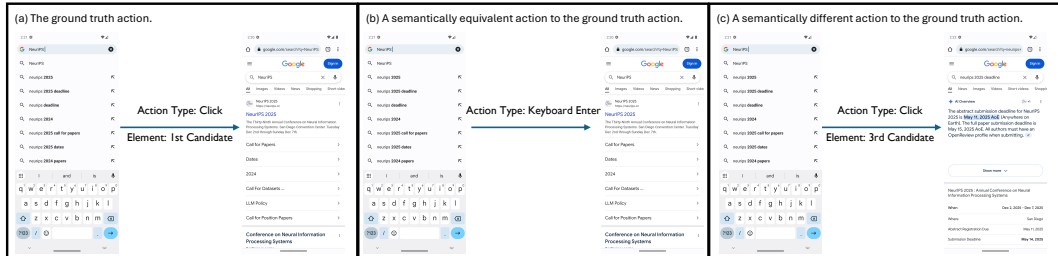

Figure 2: Examples of semantically equivalent actions: (a) and (b) lead to the same GUI state, while (c) results in a different outcome.

textual information like UI element trees (Li et al., 2024). Let $\mathcal{D}_{train} = \{(g_i, h_i, o_i, a_i^*)\}_{i=1}^N$ denote a training dataset with $N$ step samples, where $a_i^*$ is the ground truth action for sample $i$. Our objective is to fine-tune an App agent to maximise the success rate of completing the user goal $g$, while also considering generalisation and robustness. For simplicity, we assume $\gamma = 1$. The utility function $\mathcal{R}$ returns 1 when the user goal can be completed in the required maximum action steps, and 0 otherwise.

**OOD vulnerability for syntax learning.** Current supervised approaches for training App agents typically employ a syntax learning paradigm, where models learn to reproduce exactly the ground truth action string (Rawles et al., 2024a; Wang et al., 2024a; Zhang et al., 2023; Li et al., 2024). Mathematically, their learning objective can be formulated as:

$$\arg\min_\theta \mathbb{E}_{(g_i, h_i, o_i, a_i^*) \sim \mathcal{D}_{train}}[-\ell_\theta^{SFT}(g_i, h_i, o_i, a_i^*)], \tag{1}$$

where $\ell_\theta^{SFT}(g_i, h_i, o_i, a_i^*) = -\log p_\theta(a_i^* \mid g_i, h_i, o_i)$ is the supervised fine-tuning (SFT) loss function for the data sample $i$ and $p_\theta(a_i^* | g_i, h_i, o_i)$ is the likelihood of the ground truth action under the App agent policy $\pi_\theta$ for the given inputs. The key limitation of this paradigm is that it rewards the model only when the output exactly matches the ground truth action, while penalising all other outputs, even those that share identical semantics with the ground truth action. Consequently, the model optimises for token-level cross-entropy rather than understanding action semantics, leading to severe out-of-distribution (OOD) vulnerability: even minor and naturally occurring variations in UIs can be treated as OOD, causing the model to overfit to training patterns and degrade in generalisation. We formulate this OOD vulnerability as Theorem 3.1 with the proof in the Appendix.

**Theorem 3.1.** *Let $\pi_\theta^{\mathrm{SFT}}$ be an App agent trained under the syntax-learning objective of Eq. (1); for an $n$-step task with ground truth actions $\{a_1^*, \ldots, a_n^*\}$, define $P(success) = \Pr[\text{the agent completes all } n \text{ steps correctly}]$ and, for any semantically preserving but syntactically non-trivial perturbation $\delta$ satisfying $\delta(a_i^*) \neq a_i^*$ for at least one $i$, define $P(success \mid \delta) = \Pr[\text{the agent completes all } n \text{ steps correctly when each } a_i^* \text{ is replaced by } \delta(a_i^*)]$; letting $\Delta(\delta) = P(success) - P(success \mid \delta)$, we have $\Delta(\delta) > 0$.*

### 3.2 ACTION SEMANTICS LEARNING

To solve this OOD vulnerability, we introduce Action Semantics Learning (ASL), where the App agents aim to learn the semantics of ground truth actions, as displayed in Fig. 1.

In this subsection, we begin by formalising the concept of action semantics for App agents. Building on this concept, we derive a semantics-aware loss function that guides the agent to learn the semantics of ground truth actions. Finally, we introduce the semantics-aware fine-tuning pipeline of our ASL framework, which leverages the semantics-aware loss function to train App agents.

**Action semantics.** Inspired by the denotational semantics in programming language theory, which defines program semantics by its effect on system state (Stoy, 1981), we mathematically define the action semantics for App agents as the following:

**Definition 3.1. (Action Semantic).** Let $\mathcal{S}$ be the set of concrete smartphone states and $\mathcal{A}$ the set of finite actions. Moreover, $\mathcal{T} : \mathcal{S} \times \mathcal{A} \to \Delta(\mathcal{S})$ denotes the state transition function mapping a state-action pair to a probability distribution over next states. $S(a) := \mathcal{T}(s, a)$.

Intuitively, the semantics of an action generated by the App agents is defined as the *state transition* induced by the action in the UI environment. This definition is supported by the observation that, in

interactive systems such as mobile UIs, the meaning of an action is not fully captured by the action string, but rather by the transformation it causes in the environment. For instance, tapping different buttons may lead to the same screen, while actions with similar strings may have vastly different effects depending on the context, as shown in Fig. 2. Therefore, a semantic understanding of action must go beyond symbolic identifiers and focus on the consequences of the action.

**Semantics-aware loss function.** With the Definition 3.1, for a ground truth action $a_i^\star$, we define its semantic equivalence set as $\mathcal{A}_{a_i^\star}^{eq} = \{a \in \mathcal{A} \mid S(a) = S(a_i^\star)\}$. Instead of forcing the App agent to reproduce the exact $a_i^\star$, we encourage the model to assign high probability to all actions within $\mathcal{A}_{a_i^\star}^{eq}$. In real-world scenarios, $\mathcal{A}_{a_i^\star}^{eq}$ is typically unavailable, as identifying all semantically equivalent actions requires exhaustive human annotation and environment interaction. To address this, we formulate the learning objective in terms of a semantic reward:

$$J(\theta) = \mathbb{E}_{(g_i,h_i,o_i,a_i^*)\sim\mathcal{D}_{train}}\Big[\mathbb{E}_{a\sim\pi_\theta(\cdot|g_i,h_i,o_i)}\big[r(a,a_i^\star,g_i,h_i,o_i)\big]\Big], \tag{2}$$

where $r(\cdot)$ is a reward model that quantifies the probability that action $a$ is semantically equivalent to $a_i^\star$ given the input $(g_i,h_i,o_i)$. In practice, we employ a training-free reward model. We elaborate on the design of this reward model in Sec. 3.3.

To optimise the objective function in Eq. (2), we leverage the REINFORCE algorithm (Williams, 1992). This yields the following gradient estimator:

$$\nabla_\theta J(\theta) = \mathbb{E}_{(g_i,h_i,o_i,a_i^*)\sim\mathcal{D}_{train}}\Big[\mathbb{E}_{a\sim\pi_\theta(\cdot|g_i,h_i,o_i)}\big[r(a,a_i^\star,g_i,h_i,o_i)\,\nabla_\theta\log p_\theta(a\mid g_i,h_i,o_i)\big]\Big],$$

where $p_\theta(a|g_i,h_i,o_i)$ is the likelihood of an action under the App agent $\pi_\theta$ for the given inputs. Then, we design the semantics-aware loss function for a data sample $i$ as:

$$\ell_\theta^S(g_i,h_i,o_i,a_i^*) = -\mathbb{E}_{a\sim\pi_\theta(\cdot|g_i,h_i,o_i)}\big[r(a,a_i^\star,g_i,h_i,o_i)\,\log p_\theta(a\mid g_i,h_i,o_i)\big]. \tag{3}$$

In Sec. 3.4, we show that minimising the semantics-aware loss function can lead to an App agent that is more robust to the OOD vulnerability than the one fine-tuned with the syntax learning paradigm.

**Supervised fine-tuning with SEE.** Optimising the semantics-aware loss function in Eq. (3) enables App agents to learn action semantics by rewarding semantically equivalent actions. However, computing this loss requires sampling actions from the agent and scoring these actions using a reward model, which can lead to prohibitively high training costs.

To maintain efficiency, inspired by previous works in the literature (Nakano et al., 2021; Ouyang et al., 2022; Rafailov et al., 2023), we limit the number of sampled actions as one per data sample. However, sampling only a single action can lead to vanishing gradients when the sampled action is not semantically equivalent to the ground truth, hindering effective learning and potentially leading to sub-optimal performance. To address this, we propose a combined loss function that integrates our semantics-aware loss function with the SFT loss function. The loss function is formulated as follows:

$$\ell_\theta^{ASL}(g_i,h_i,o_i,a_i^*) = \ell_\theta^S(g_i,h_i,o_i,a_i^*) + \lambda_i\,\ell_\theta^{SFT}(g_i,h_i,o_i,a_i^*), \tag{4}$$

where $\lambda_i = \big(1-(1-\alpha_i)\,r(a,a_i^\star,g_i,h_i,o_i)\big)$, and $\alpha_i = \exp\big(-\ell_\theta^{SFT}(g_i,h_i,o_i,a_i^*)\big)$. We use $\alpha_i$ to quantify the syntax dissimilarity between the action generated by the App agent and the ground truth action. The SFT loss function value used to compute $\alpha_i$ is detached from the gradient computational graph, ensuring it only modulates the SFT term without creating unintended gradient interactions.

Intuitively, the combined loss function operates as follows: (1) when the sampled action is semantically correct but syntactically different, learning is primarily guided by the semantic-aware loss; (2) when the action matches both semantics and syntax, both loss components contribute jointly; and (3) when the action matches neither, the SFT loss dominates, ensuring stable gradient flow even in the absence of semantic reward. This design efficiently mitigates gradient vanishing issues, as it requires sampling only a single action from the App agent per training instance.

**Reinforcement fine-tuning with SEE.** Beyond supervised fine-tuning, we follow DigiRL (Bai et al., 2024) and optimise the policy with advantage-weighted regression (AWR) (Peng et al., 2019). For each transition $(g_i,h_i,o_i,a_i)$, we inject a thresholded semantic reward $\tilde{r}(a_i,\tilde{a}_i,g_i,h_i,o_i) = \mathbb{1}\{r(a_i,\tilde{a}_i,g_i,h_i,o_i) > \tau\}\,r(a_i,\tilde{a}_i,g_i,h_i,o_i)$, where $\tilde{a}_i$ is the teacher model-predicted reference action and $\tau$ is a semantic threshold. We then define the augmented reward $r^{\text{see}}(\cdot)$ as:

$$r^{\text{see}}(a_i,\tilde{a}_i,g_i,h_i,o_i) = r^{\text{env}}(a_i,g_i,h_i,o_i) + \beta\,\tilde{r}(a_i,\tilde{a}_i,g_i,h_i,o_i), \tag{5}$$

where $r^{\mathrm{env}}(\cdot)$ is the original environment reward, $\tilde{r}(\cdot)$ is the SEE-based semantic reward, and $\beta > 0$ adjusts the weight of the semantic reward. The step-level advantage is computed as:

$$
\begin{aligned}
A^{\mathrm{DigiRL\text{-}SEE}}(g_i, h_i, o_i, a_i, \tilde{a}_i) = {} & \eta^{H-i} \, r^{\mathrm{env}}(a_H, g_H, h_H, o_H) \\
& + \left(1 - \eta^{H-i}\right)\!\left(V^{\mathrm{step}}(g_{i+1}, h_{i+1}, o_{i+1}) + r^{\mathrm{see}}(a_i, \tilde{a}_i, g_i, h_i, o_i) - V^{\mathrm{step}}(g_i, h_i, o_i)\right),
\end{aligned}
\tag{6}
$$

where $H$ is the horizon, $\eta \in [0,1]$ balances the high-variance Monte Carlo estimate $\eta^{H-i} \, r^{\mathrm{env}}(a_H, g_H, h_H, o_H)$ and the value-based baseline from a learned doubly-rubost estimator $V^{\mathrm{step}}$. The optimisation objective of the policy $\pi_\theta$ is a hard-filtered AWR loss as:

$$
\ell_\theta^{\mathrm{DigiRL\text{-}SEE}}(g_i, h_i, o_i, a_i, \tilde{a}_i) = -\mathbb{1}\!\left\{A^{\mathrm{DigiRL\text{-}SEE}}(g_i, h_i, o_i, a_i, \tilde{a}_i) > 0\right\} \log p_\theta(a_i \mid g_i, h_i, o_i).
\tag{7}
$$

SEE augments DigiRL with step-level semantic feedback, thereby improving credit assignment under sparse, binary outcomes and yielding stronger generalisation on long-horizon UI control. In the next section, we elaborate on the Semantic Estimator (SEE) for App agents.

### 3.3 SEE: SEMANTIC ESTIMATOR

To define the semantics of an action, we focus on its effect rather than its string syntax or associated metadata. Predefined actions often come in varying formats, and typically include parameters such as the ID of a UI element that lacks standalone semantic meaning, especially without the associated user input or context. In contrast, we propose that the result of executing an action within a given state carries rich semantic information, and that this outcome can serve as a more meaningful definition of the action itself. We introduce a SEmantic Estimator (SEE) that consists of two key components: (1) a world model that simulates the smartphone UI state transitions, and (2) a semantic calculator that estimates the semantic reward by quantifying the similarity between the transitions.

**World model.** Inspired by recent advances in world-model–based web agents (Chae et al., 2024), which leverage predicted future observations as rewards for prompt engineering, SEE utilises a world model to mimic the state transition function $\mathcal{T}(s, a)$ mentioned in Definition 3.1. Mathematically, the workflow of this world model can be formulated as:

$$
\Delta \hat{o}_t = \Phi(o_t, a_t),
\tag{8}
$$

where $o_t$ is the textual description of the UI state $s_t$, $a_t$ is the textual description of an action, and $\Delta \hat{o}_t$ is the description of the impact of the action $a_t$ on the UI state $s_t$. In practice, we implement this world model based on the proprietary LLM API by leveraging the LLM's strong capabilities in language understanding and world knowledge to simulate how the environment would evolve in response to the action. The prompt of our world model can be found in the Appendix H.

**Semantic calculator.** With the state transitions generated by the world model, SEE employs a semantic calculator to generate the semantic reward for our ASL framework-generated transitions in two steps. Firstly, the semantic calculator uses BERT (Devlin et al., 2019) to project the generated textual description for state transitions into a feature space. This process can be formulated as:

$$
f_t^{gt} = \mathrm{BERT}(\Delta \hat{o}_t^{gt}), \quad f_t^{pre} = \mathrm{BERT}(\Delta \hat{o}_t^{pre}),
\tag{9}
$$

where $\Delta \hat{o}_t^{gt}$ and $\Delta \hat{o}_t^{pre}$ denote the textual descriptions of the ground truth and predicted state transitions at timestep $t$, respectively, and $f_t^{gt}, f_t^{pre} \in \mathbb{R}^d$ are their corresponding feature representations. Then, SEE computes the cosine similarity between the feature vectors to measure semantic alignment:

$$
cos_t = \frac{\langle f_t^{gt}, f_t^{pre} \rangle}{|f_t^{gt}| \cdot |f_t^{pre}|}.
\tag{10}
$$

This similarity score is used in our ASL framework as the semantic reward. A higher similarity score indicates a higher probability that a semantic equivalence exists between the action generated by the App agent and the ground truth action, providing a reward signal that guides the training of the agent towards semantically meaningful actions, even when the syntax may differ.

### 3.4 THEORETICAL ANALYSIS

In this section, we demonstrate that enabling the App agent to learn action semantics by minimising the semantic loss function defined in Eq. (3) leads to an agent that is more robust to the OOD

problems compared to an agent that purely optimises the learning objective formulated in Eq. (1). Mathematically, we present Theorem 3.2 with the proof in the Appendix.

**Theorem 3.2.** *Let $\pi_\theta^{\mathrm{S}}$ be an App agent trained by minimising the semantics-aware loss function defined in Eq. (3), and let $\pi_\theta^{\mathrm{SFT}}$ be an App agent trained under the syntax-learning objective defined in Eq. (1). For an $n$-step task with ground truth actions $\{a_1^*, \ldots, a_n^*\}$, consider any semantically preserving but syntactically non-trivial perturbation $\delta$ such that $\delta(a_i^*) \neq a_i^*$ for at least one $i$. Define $P^{\mathrm{S}}(success \mid \delta)$ as the success rate of $\pi_\theta^{\mathrm{S}}$ and $P^{\mathrm{SFT}}(success \mid \delta)$ as the success rate of $\pi_\theta^{\mathrm{SFT}}$ under the perturbation $\delta$. Then, we have $P^{\mathrm{S}}(success \mid \delta) - P^{\mathrm{SFT}}(success \mid \delta) > 0$.*

This theorem highlights that, by focusing on the semantic equivalence of actions rather than exact syntactic matches, the App agent trained with Eq. (3) exhibits higher success rates than an agent trained solely on the syntax-learning objective when encountering the OOD data. We further provide experimental support for our learning framework in Sec. 4.

## 4 EXPERIMENTS

In this section, we first introduce our experimental setup. We then demonstrate the practical value of the proposed ASL framework on two online smartphone operation Apps benchmarks. To evaluate the robustness of the agent fine-tuned by our ASL framework, we also use an offline smartphone operation benchmark. Additionally, we interpret our semantic similarity score as a step-wise reward within an online RL fine-tuning pipeline, and further demonstrate its effectiveness. Finally, we conduct an ablation study to study the contribution of components within the ASL framework.

### 4.1 EXPERIMENTAL SETUP

**Datasets.** We conducted our experiments on five benchmark datasets: *AndroidLab (Xu et al., 2024a)*, *AndroidWorld (Rawles et al., 2024a)*, *AndroidControl (Li et al., 2024)*, *Android-in-the-Wild (AitW) (Rawles et al., 2024b)*, and *WebArena-Lite* (Zhou et al., 2024a; Liu et al., 2025a). These datasets encompass a diverse range of mobile UI settings and user interaction tasks, enabling a comprehensive evaluation of our model's generalisation and understanding capabilities.

**Metrics.** For the online datasets such as *AndroidWorld*, performance is measured by the task-level success rate (TSR), i.e., the ratio of completed tasks to total tasks, where a task is successful if finished within the dataset's step limit. For the offline dataset *AndroidControl*, we use the step-level success rate (SSR), i.e., the ratio of correctly predicted steps to total steps, where a step is successful if the generated action matches the ground truth

**Implementation details.** For the semantic evaluator, we use Gemini-Flash-2.0 (Hassabis & Kavukcuoglu, 2024) as the LLM to predict the change that an action will bring to the UI state. Batch size was set to 1 and trained for 3 epochs. The training dataset includes 2K human-annotated step-wise samples from apps in the AndroidWorld emulator environment and 70K samples from the AndroidControl benchmark. The 2K human-generated samples are curated to ensure no overlap with the test set in AndroidControl. Low-level instructions from AndroidControl are not used. T3A* refers to our prompt adapted from T3A, detailed in the Appendix. Qwen-2.5-7B-Instruct-ASL refers to the model fine-tuned with our ASL loss, as defined in Eq. 4.

We report experiments on the AndroidWorld, AndroidLab, and AndroidControl datasets using ASL fine-tuned Qwen-2.5-7B. For the AitW tasks and WebArena-Lite benchmark, we follow Eq. 7 and integrate our SEE module into their RL fine-tuning framework.

### 4.2 ONLINE DATASET COMPARISON

In this experiment, we evaluated model performance in real-time interactive environments using the AndroidWorld (Rawles et al., 2024a) and AndroidLab (Xu et al., 2024a) datasets. This setup is well-suited for testing our core motivation: that models should go beyond syntactic patterns and learn the semantics of actions—that is, their actual outcomes. Specifically, AndroidWorld provides an emulator that mimics real-world conditions by automatically executing actions and employs a hand-crafted rule-based system to verify task completion by checking system status. For AndroidLab, we use 30 provided tasks, excluding those involving question answering. We evaluate AndroidLab by

Table 1: Comparison of different agents in terms of success rate in the AndroidWorld environment. T3A* refers to our prompt adapted from T3A, presented in the Appendix.

| Type | Agent | Model | Input Modality | TSR ↑ | | | Overall TSR ↑ |
|------|-------|-------|----------------|-------|---|---|---------------|
| | | | | Easy | Medium | Hard | |
| Prompt-Driven | SeeAct | GPT-4o | Image + Text | 34.2 | 15.5 | 4.2 | 22.0 |
| | T3A | GPT-4o | Image | 64.9 | 26.2 | **14.6** | 41.9 |
| | M3A | GPT-4o | Image + Text | 60.5 | 20.2 | 8.3 | 36.6 |
| | T3A* | Qwen2.5-7B-Instruct | Text | 5.6 | 0.0 | 0.0 | 2.5 |
| Fine-Tuned | T3A* | Qwen2.5-7B-Instruct-SFT | Text | **72.2** | 26.8 | 3.13 | 42.5 |
| | | **Qwen2.5-7B-Instruct-ASL (Ours)** | Text | **72.2** | **32.1** | 12.5 | **46.3** |

Table 2: Comparison of different agents in terms of success rate in the AndroidLab environment.

| Type | Agent | Model | Input Modality | TSR |
|------|-------|-------|----------------|-----|
| Prompt-Driven | AndroindLab-Agent | Claude-3.5-Sonnet | Image + Text | 24.2 |
| | | Gemini-1.5-Pro | Image + Text | 13.0 |
| | | GPT-4o | Image + Text | 30.1 |
| | | Gemini-1.5-Pro | Text | 17.2 |
| | | GPT-4o | Text | 24.7 |
| Fine-Tuned | T3A* | Qwen-2.5-7B-Instruct-SFT | Text | 33.3 |
| | | **Qwen-2.5-7B-Instruct-ASL (Ours)** | Text | **35.5** |

executing actions within the AndroidWorld emulator and using an LLM-based evaluator to determine whether each task is completed successfully. Table 1 presents the results on AndroidWorld, reporting success rates across tasks of varying difficulty—categorised as easy, medium, and hard—as well as the overall success rate. As shown in the table, our method outperforms the baseline by a significant margin of 3.8% in overall success rate. The most substantial improvements are observed in medium and hard tasks. We further evaluate our method on the AndroidLab benchmark, with superior results presented in Table 2. Together, these results highlight the generalisation capability of our model in real-world mobile UI environments and underscore the effectiveness of our ASL framework.

### 4.3 OFFLINE DATASET COMPARISON

In this experiment, we evaluated model performance on the offline benchmark AndroidControl (Li et al., 2024). Specifically, we compared predicted actions against ground truth annotations. For action types, we assess whether the predicted action type string exactly matches the ground truth. For actions that involve coordinates, such as click and long press, we follow the evaluation protocol (Rawles et al., 2024b) to determine whether the ground truth coordinate falls within the bounding box of the predicted UI element. As shown in Table 3, we collect and evaluate both prompt-driven and models fine-tuned on AndroidControl. Our model achieves the highest performance across all four splits, highlighting the model's robustness to new tasks, categories, and applications.

### 4.4 RL FINE-TUNING WITH SEE

To further validate the effectiveness, we evaluate our SEE module with different RL training pipelines on the AitW tasks (Rawles et al., 2024b) and WebArena-Lite benchmark (Zhou et al., 2024a; Liu et al., 2025a). Following prior work (Bai et al., 2024; Wu et al., 2025), we train on the full AitW training task split and evaluate on the first 96 tasks from both train and test splits of the General and Web Shopping categories, with success judged by Gemini (Team et al., 2024). For WebArena-Lite, all methods are trained and evaluated on all 165 tasks, with success assessed by a pretrained outcome reward model. Additional details are provided in Appendix E.2. As shown in Table 4 and Table 5, both baselines consistently benefit from our SEE module. On AitW, Filtered BC-SEE outperforms Filtered BC by 6.9% on General train, while DigiRL-SEE achieves the best overall performance, surpassing DigiRL by 5.9% and 4.2% on General train and test. On WebArena-Lite, Filtered BC-SEE improves by 2.5%, and DigiRL-SEE further outperforms DigiRL by 1.8%. These results indicate that our SEE module provides denser training signals, alleviates the sparse reward issue, and enhances generalisation across mobile app and web control tasks.

Table 3: Comparison across agents of step success rate in the four splits of AndroidControl.

| Type | Agent | Model | Input Modality | SSR ↑ | | | |
|---|---|---|---|---|---|---|---|
| | | | | IDD | Task-Unseen | Cat-Unseen | App-Unseen |
| Prompt-Driven | SeeAct | Gemini-2.5-Flash | Image + Text | 36.7 | 35.9 | 33.7 | 34.4 |
| | | GPT-4o | | 31.5 | 30.7 | 30.6 | 30.9 |
| | M3A | Gemini-2.5-Flash | | 49.4 | 47.9 | 47.9 | 47.9 |
| | | GPT-4o | | 60.8 | 59.3 | 60.8 | 60.4 |
| | T3A | Gemini-2.5-Flash | Text | 49.1 | 45.4 | 44.4 | 45.0 |
| | | GPT-4o | | 56.1 | 55.8 | 56.5 | 54.2 |
| | T3A* | Gemini-2.5-Flash | | 46.8 | 45.3 | 44.7 | 43.5 |
| | | Qwen2.5-7B-Instruct | | 26.1 | 27.2 | 26.7 | 26.3 |
| Fine-Tuned | T3A* | Qwen2.5-7B-Instruct-SFT | Text | 67.4 | 61.2 | 59.7 | 59.7 |
| | | **Qwen2.5-7B-Instruct-ASL (Ours)** | Text | **69.4** | **62.3** | **61.2** | **61.8** |

Table 4: Evaluation of our SEE module on the AitW General and Web Shopping tasks.

| Task | | AppAgent | CogAgent | AutoUI | Filtered BC | Filtered BC-SEE | DigiRL | DigiRL-SEE |
|---|---|---|---|---|---|---|---|---|
| General | Train | 14.6% | 25.0% | 12.5% | 53.5% | 60.4% (+6.9%) | 64.9% | **70.8%**(+5.9%) |
| | Test | 16.7% | 25.0% | 14.6% | 62.5% | 63.5% (+1.0%) | 67.7% | **71.9%**(+4.2%) |
| Web Shopping | Train | 5.2% | 31.3% | 14.6% | 53.6% | 55.9%(+2.3%) | 55.3% | **57.3%**(+2.0%) |
| | Test | 8.3% | 38.5% | 17.7% | 54.2% | **56.3%** (+2.1%) | 41.3% | 55.2% (+13.9%) |

Table 5: Evaluation of our SEE module on the WebArena-Lite benchmark.

| | SFT | Filtered BC | Filtered BC-SEE | DigiRL | DigiRL-SEE |
|---|---|---|---|---|---|
| Task SSR | 20.0% | 23.0% | 25.5% (+2.5%) | 30.3% | **32.1%** (+1.8%) |

Table 6: Ablation study on the loss function.

| | $\ell_\theta^{SFT}$ | $\lambda_i \ell_\theta^{SFT}$ | $\ell_\theta^{ASL}$ |
|---|---|---|---|
| OOD SSR (%) | 49.8 | 49.9 | **51.7** |

Table 7: Ablation study on reward design.

| | Gemini-2.0-Flash | SEE (Ours) |
|---|---|---|
| OOD SSR (%) | 50.6 | **51.7** |

## 4.5 ABLATION STUDY

In this section, we study two key components of our ASL framework: the loss function and the reward model used during training. Ablation studies are conducted using the Qwen2.5-7B-Instruct model, trained on 3K samples from the AndroidControl training set and evaluated on 1K samples from the AndroidControl OOD test set. Results for the loss function ablation are presented in Table 6, where $\ell_\theta^{\text{SFT}}$ denotes the standard SFT loss used in the syntax learning paradigm, $\lambda_i \ell_\theta^{\text{SFT}}$ represents the weighted SFT loss in our framework, and $\ell_\theta^{\text{ASL}}$ refers to the full loss function used in ASL. The results demonstrate that $\ell_\theta^{\text{ASL}}$ leads to the highest performance, highlighting its effectiveness in enabling the App agent to learn action semantics and improving robustness to OOD scenarios.

Table 7 reports the ablation results on the reward model. In this table, 'Gemini-2.0-Flash' denotes a baseline that directly uses Gemini-2.0-Flash to compute the semantics reward, while SEE refers to our proposed semantics estimator. The superior performance of SEE confirms its benefit in providing more informative and stable reward signals during training. These findings validate the effectiveness of both the loss function and the SEE reward model as core components of the ASL framework.

## 5 CONCLUSION

We present Action Semantics Learning (ASL), a novel framework for training App agents that prioritises semantic understanding of actions rather than syntactic reproduction. By defining action semantics as the UI state transition induced by an action, and introducing a lightweight semantic estimator (SEE) to provide reward signals during training, ASL enables agents to treat semantically equivalent actions equally, even when their textual forms differ.

We theoretically prove that ASL improves robustness against out-of-distribution (OOD) variations. In online evaluation on AndroidWorld and AndroidLab, ASL-trained agents outperform both prompt-based closed-source agents and fine-tuned baselines, achieving higher task success rates, especially on medium and hard tasks that require flexible reasoning and semantic generalisation. On the offline AndroidControl benchmark, ASL significantly improves step-level accuracy under unseen task and app settings compared to syntax-based fine-tuning. Experiments applying our SEE reward to RL fine-tuning pipelines further demonstrate the superiority of our approach.

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

## A STATEMENT ON LLM USAGE

We disclose that large language model (LLM) tools were used solely for language refinement, including improving grammar and polishing phrasing. LLMs were not used to generate scientific content, research ideas, experiment designs, data, analyses, or code. All suggestions and modifications from these tools were made under the direct supervision and final approval of the authors, and all authors are fully aware of and consent to this usage.

## B EXPERIMENTAL ENVIRONMENT

For the experiments of Online Dataset Comparison and Offline Dataset Comparison, all training is conducted on NVIDIA V100 GPUs. Our online tests follow the instructions of the original benchmark datasets. Specifically, we evaluate our App agent on the Android Virtual Device (AVD) with system images Android 13.0 Tiramisu (API Level 33) following the setup of prior work (Rawles et al., 2024a; Li et al., 2024; Xu et al., 2024a).

For the experiment of RL Fine-tuning with SEE, each run on the AitW tasks with Filtered BC-SEE and DigiRL-SEE takes approximately 24 hours on a single NVIDIA GeForce RTX 4090 GPU and 8 Intel Xeon CPUs. The online interaction and evaluation is conducted on the AVD with system images Android 9.0 Pie (API Level 28), following the setup of prior work (Bai et al., 2024; Wu et al., 2025). On the WebArena-Lite benchmark, each run of Filtered BC-SEE, DigiRL-SEE, and WebRL-SEE requires about 24 hours on 8 NVIDIA GeForce RTX 4090 GPUs and 8 Intel Xeon CPUs.

## C PROOF FOR THEOREM 3.1

*Proof.* For the App agent trained by minimising the syntax learning objective, we have:

$$P(\text{success}) = \prod_{i=1}^{n} p_\theta\big(a_i^* \mid g_i, h_i, o_i\big) = \gamma^n,$$

where $\gamma \to 1$. Let $\tilde{o}_i = \delta(o_i)$ be the perturbed observation that never appears in the training set, which makes the correct action satisfying $\delta(a_i^*) \neq a_i^*$. Since the optimiser has no incentive to allocate *more* probability to any specific action under $\tilde{o}_i$ than under $o_i$, we can have:

$$p_\theta\big(\delta(a_i^*) \mid g_i, h_i, \tilde{o}_i\big) \leq p_\theta\big(\delta(a_i^*) \mid g_i, h_i, o_i\big).$$

Theron, we have:

$$P(\text{success} \mid \delta) = \prod_{i=1}^{n} p_\theta\big(\delta(a_i^*) \mid g_i, h_i, \tilde{o}_i\big) \leq \prod_{i=1}^{n} p_\theta\big(\delta(a_i^*) \mid g_i, h_i, o_i\big) = (1 - \gamma - \eta)^n,$$

where $\eta \in (0, 1 - \gamma)$. Since $\gamma \to 1$, we have $\gamma > (1 - \gamma - \eta)$. $P(\text{success}) > P(\text{success} \mid \delta)$ and $\Delta(\delta) = P(\text{success}) - P(\text{success} \mid \delta) > 0$ as claimed. $\square$

## D PROOF FOR THEOREM 3.2

*Proof.* Let $\mathcal{A}_{a_i^\star}^{eq}$ be the semantics equivalent space for the ground-truth action $a_i^\star$. Moreover, since the given perturbation is a semantically preserving but syntactically non-trivial perturbation, we have: $\forall i$, $|\mathcal{A}_{a_i^\star}^{eq}| \geq 2$. Assume that the semantics-aware agent $\pi_\theta^{\mathrm{S}}$ achieves an expected reward within $\varepsilon$ of the optimal, i.e., $J(\theta) \geq J^* - \varepsilon$, where $J^*$ is the maximum achievable reward. By Markov's inequality, the probability that the agent selects an action with reward less than $1 - \sqrt{\varepsilon}$ is at most $\sqrt{\varepsilon}$. Therefore, for each step $i$, the probability of selecting a semantically equivalent action under perturbation $\delta$ is at least $1 - \sqrt{\varepsilon}$, where $\sqrt{\varepsilon} \to \frac{1}{|\mathcal{A}_{a_i^\star}^{eq}|} \leq \frac{1}{2}$. Assuming independence across steps, we can have:

$$P^{\mathrm{S}}(\text{success} \mid \delta) \geq (1 - \sqrt{\varepsilon})^n.$$

On the other hand, the syntax-only agent $\pi_\theta^{\mathrm{SFT}}$, trained to minimise cross-entropy loss, assigns $\gamma \to 1$ to the exact training action $a_i^*$ and low probability to other actions. Under perturbation $\delta$, where

$\delta(a_i^*) \neq a_i^*$, the agent's probability of selecting the correct action decreases to at most $1 - \gamma$ per step. Assuming independence across steps, we can have:

$$P^{\text{SFT}}(\text{success} \mid \delta) \leq (1 - \gamma)^n.$$

Given that $0 < \sqrt{\varepsilon} < \gamma < 1$, it follows that $(1 - \sqrt{\varepsilon})^n > (1 - \gamma)^n$, implying that $P^{\text{S}}(\text{success} \mid \delta) > P^{\text{SFT}}(\text{success} \mid \delta)$. Therefore, $P^{\text{S}}(\text{success} \mid \delta) - P^{\text{SFT}}(\text{success} \mid \delta) > 0$. $\qquad \square$

# E    IMPLEMENTATION DETAILS

## E.1    DATASET

*AndroidWorld* includes a high-fidelity simulator that emulates realistic Android device behaviour, enabling interactions with 20 real-world applications across 116 user-defined tasks. To determine whether a task is successfully completed, each task is paired with manually authored rules that provide precise success criteria. *AndroidLab* includes 138 predefined tasks across nine Android applications built on virtual devices. Each task is divided into multiple required page states as sub-goals, with UI tree structure matching verifying correct traversal. The AndroidLab environment ensures reproducibility and eliminates external network or time dependencies. *AndroidControl* comprises 15,283 human demonstrations spanning 14,548 unique tasks across 833 Android applications. *Android in the Wild (AitW)* is a large-scale dataset containing 715k human demonstrations across 30k unique natural language instructions and 357 apps, capturing realistic multi-step interactions on Android devices and websites. *WebArena-Lite* is a human-verified subset from WebArena, which contains 165 tasks spanning diverse websites such as Gitlab, maps, forums, shopping, and CMS, designed to evaluate multimodal web agents under realistic and complex user instructions .

## E.2    RL FINE-TUNING

Supervised fine-tuning uses human-annotated ground-truth actions to compute the semantic score between the predicted and ground-truth actions. In contrast, for RL fine-tuning, no human annotations are available. Instead, we prompt Gemini (Comanici et al., 2025) as a teacher model to predict an action and adopt it as the reference action, since Gemini demonstrates high semantic fidelity and robustness, making it a suitable proxy for human-annotated actions.

In the experiments on AitW tasks, we evaluate the effect of our SEE module against various methods, including AppAgent (Zhang et al., 2025), CogAgent (Hong et al., 2024), AutoUI (Zhang & Zhang, 2024), Filtered BC (Pan et al., 2024), and DigiRL (Bai et al., 2024). For Filtered BC, Filtered BC-SEE, DigiRL, and DigiRL-SEE, all experiments are initialised from the AutoUI-Base model, ensuring a consistent starting point across methods. In the experiments on the WebArena-Lite benchmark, the compared methods include SFT, Filtered BC, DigiRL, and our SEE-augmented counterparts. All models are initialised from the SFTed Llama-3.1-8B model released by WebRL (Qi et al., 2025), which was trained on the trajectory data collected from the WebArena-Lite benchmark.

For the implementation of Filtered BC-SEE, we filters out the successful trajectories $\mathcal{D}_{\text{succ}}$ and applies behavior cloning to the ground-truth action $a_i^\star$. We then extend the filtered BC loss with a semantics-aware term:

$$\ell_\theta^S(g_i, h_i, o_i, \tilde{a}_i) = -\tilde{r}(a_i, \tilde{a}_i, g_i, h_i, o_i) \log p_\theta(a_i \mid g_i, h_i, o_i), \tag{11}$$

and the loss function of Filtered BC-SEE is:

$$\ell_\theta^{\text{FBC-SEE}}(g_i, h_i, o_i, a_i^\star, \tilde{a}_i) = \ell_\theta^S(g_i, h_i, o_i, \tilde{a}_i) + \lambda_i \, \ell_\theta^{SFT}(g_i, h_i, o_i, a_i^\star), \tag{12}$$

where the original filtered BC loss $\ell_\theta^{SFT}(g_i, h_i, o_i, a_i^\star) = -\log p_\theta(a_i^\star \mid g_i, h_i, o_i)$, $\lambda_i = 1 - (1 - \alpha_i)\,\tilde{r}$, and $\alpha_i = \exp(-\ell_\theta^{SFT}(g_i, h_i, o_i, a_i^\star))$ is defined in Eq. 4.

The VLMs used in the experiments of RL fine-tuning are summarised in Table 8, and the hyperparameter settings are provided in Table 9 and Table 10 respectively.

# F    TRAINING CURVES

To further illustrate the effect of our SEE module, we present the training curves on the AitW General and Web Shopping tasks under both the Filtered BC and DigiRL optimisation settings in Fig. 3 and

Table 8: Summary VLMs used in RL fine-tuning.

| Component | Task | | Description |
| | AitW | WebArena-Lite | |
| --- | --- | --- | --- |
| Actor Model | AutoUI-Base | Llama-3.1-8B | Serves as the current policy for generating rollout actions. |
| Teacher Model | Gemini-2.5-Flash | Gemini-2.5-Flash | Provides proxy actions to replace costly human annotations. |
| World Model | Gemini-2.5-Flash | Gemini-2.5-Flash | Predicts the state transition given an action. |
| Evaluator | Gemini-1.5-Pro | Llama-3.1-8B | Automatically evaluates the success of tasks. |

Table 9: Hyperparameter settings for the experiments on the AitW tasks.

| Hyperparameter | Value |
| --- | --- |
| Batch Size | 4 |
| Total Trajectories | 1000 |
| Learning Rate | 1e-4 |
| Update Epoch (Actor) | 20 |
| Update Epoch (Critic) | 5 |
| Maximum Gradient Norm | 0.01 |
| Semantic Threshold ($\tau$) | 0.6 |
| Discount Factor ($\eta$) | 0.5 |
| Semantic Reward Weight ($\beta$) | 0.5 |

Fig. 4 respectively. The curves compare the baseline methods with their SEE-augmented counterparts, showing how semantic-level feedback influences convergence dynamics.

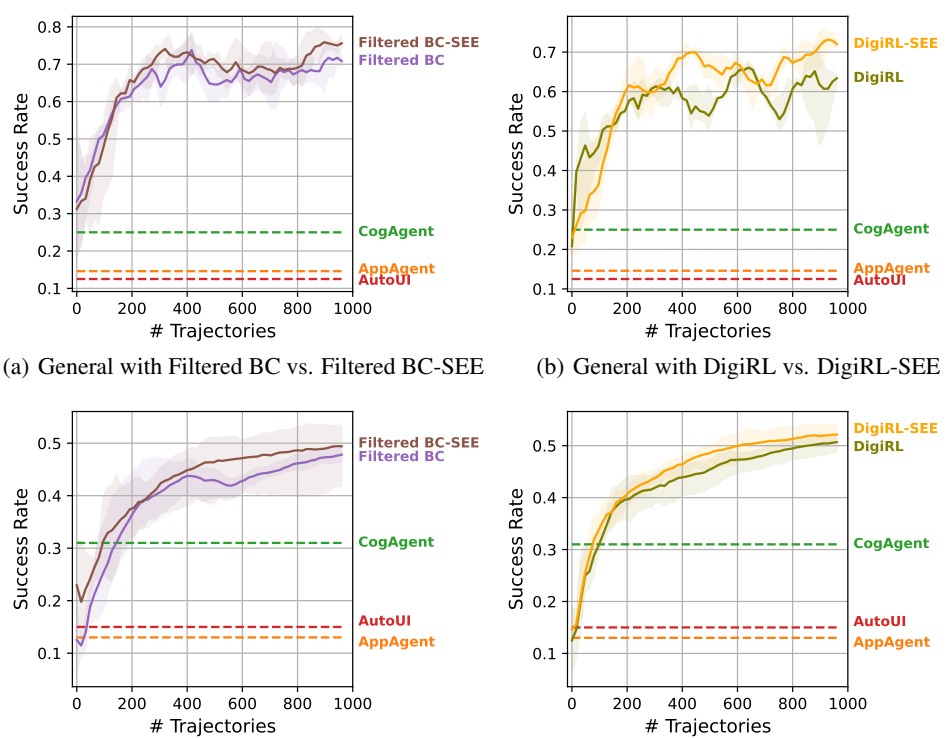

(a) General with Filtered BC vs. Filtered BC-SEE

(b) General with DigiRL vs. DigiRL-SEE

(c) Web Shopping with Filtered BC vs. Filtered BC-SEE

(d) Web Shopping with DigiRL vs. DigiRL-SEE

Figure 3: Training curves on AitW General and Web Shopping tasks. In all cases, incorporating our semantic estimator (SEE) leads to faster convergence and higher final success rates compared with the corresponding baselines.

Table 10: Hyperparameter settings for the experiments on the WebArena-Lite benchmark.

| Hyperparameter | Value |
|---|---|
| Batch Size | 128 |
| Total Trajectories | 1000 |
| Learning Rate | 1e-6 |
| Update Epoch (Actor) | 1 |
| Update Epoch (Critic) | 1 |
| Maximum Gradient Norm | 1.0 |
| Semantic Threshold ($\tau$) | 0.6 |
| Discount Factor ($\eta$) | 0.9 |
| Semantic Reward Weight ($\beta$) | 0.5 |

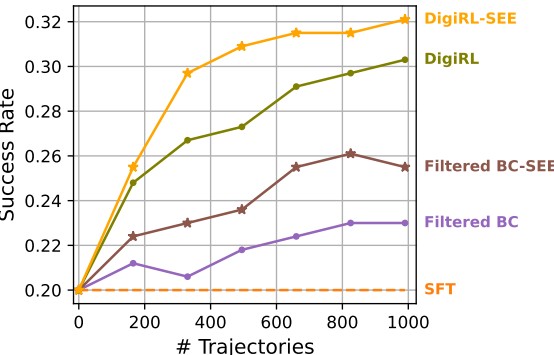

Figure 4: Training curves on the WebArena-Lite benchmark. Compared to the original baselines, incorporating our SEE leads to achieve smoother optimisation dynamics and reach higher final task success rates, further confirming the effectiveness of semantic-level feedback in web control.

## G  ADDITIONAL EXPERIMENT

In this section, we present experiments on the AndroidControl random-500 setup Li et al. (2024), comparing our method with models including Aria-UI Yang et al. (2024), GUI-R1 Luo et al. (2025), and AGUVIS Xu et al. (2024b). As shown in Table 11, our approach outperforms these models.

Table 11: Comparison on AndroidControl random 500 setup Li et al. (2024).

| Method | Step.Acc |
|---|---|
| Aria-UI | 10.2 |
| GUI-R1-3B | 46.6 |
| GUI-R1-7B | 51.7 |
| AGUVIS-7B | 61.5 |
| AGUVIS-72B | 66.4 |
| ASL-7B (Ours) | **68.9** |

# H PROMPT

## H.1 PROMPT FOR T3A*

You are an agent who can operate an Android phone on behalf of a user. Based on user's goal/request, you may
 - Answer back if the request/goal is a question (or a chat message), like user asks "What is my schedule for today?"
 - Complete some tasks described in the requests/goals by performing actions (step by step) on the phone.
When given a user request, you will try to complete it step by step. At each step, a list of descriptions for most UI elements on the current screen will be given to you (each element can be specified by an index), together with a history of what you have done in previous steps. Based on these pieces of information and the goal, you must choose to perform one of the action in the following list (action description followed by the JSON format) by outputing the action in the correct JSON format.
 - If you think the task has been completed, finish the task by using the status action with complete as goal_status: {{"action_type": "status", "goal_status": "complete"}}
 - Answer user's question:" {{"action_type": "answer", "text": "<answer_text>"}}
 - Click/tap on a UI element (specified by its index) on the screen: {{"action_type": "click", "index": <target_index>}}.
 - Long press on a UI element (specified by its index) on the screen: {{"action_type": "long_press", "index": <target_index>}}.
 - Clear the text in a text field (specified by its index): "action_type": "clear_text", "index": <target_index>}}
 - Type text into an editable text field (specified by its index), this action contains clicking the text field, typing in the text and pressing the enter, so no need to click on the target field to start: {{"action_type": "input_text", "text": <text_input>, "index": <target_index>}}
 - Press the Enter key: {{"action_type": "keyboard_enter"}}
 - Navigate to the home screen: {{"action_type": "navigate_home"}}
 - Navigate back: {{"action_type": "navigate_back"}}
 - Scroll the screen or a scrollable UI element in one of the four directions, use the same numeric index as above if you want to scroll a specific UI element, leave it empty when scroll the whole screen: {{"action_type": "scroll", "direction": <up, down, left, right>, "index": <optional_target_index>}} Notably, scroll left to reveal more on the left, scroll right to reveal more on the right, scroll up to reveal more above, and scroll down to reveal more below.
 - Open an app (nothing will happen if the app is not installed): {{"action_type": "open_app", "app_name": <name>}}
 - Wait for the screen to update: {{"action_type": "wait"}}
The overall user goal/request is: {goal}
Here is a history of what you have done so far:
{history}
Here is a list of descriptions for some UI elements on the current screen:
{ui_elements_description}
Here are some useful guidelines you need to follow:
General
 - Usually there will be multiple ways to complete a task, pick the easiest one. Also when something does not work as expected (due to various reasons), sometimes a simple retry can solve the problem, but if it doesn't (you can see that from the history), try to switch to other solutions.
 - Sometimes you may need to navigate the phone to gather information needed to complete the task, for example if user asks "what is my schedule tomorrow", then you may want to open the calendar app (using the `open_app` action), look up information there, answer user's question (using the `answer` action) and finish (using the `status` action with complete as goal_status).
 - For requests that are questions (or chat messages), remember to use the `answer` action to reply to user explicitly before finish! Merely displaying the answer on the screen is NOT sufficient (unless the goal is something like "show me ...").
 - If the desired state is already achieved (e.g., enabling Wi-Fi when it's already on), you can just complete the task.
Action Related
 - Use the `open_app` action whenever you want to open an app (nothing will happen if the app is not installed), do not use the app drawer to open an app unless all other ways have failed.
 - Use the `input_text` action whenever you want to type something (including password) instead of clicking characters on the keyboard one by one. Sometimes there is some default text in the text field you want to type in, remember to delete them before typing.
 - For `click`, `long_press` and `input_text`, the index parameter you pick must be VISIBLE in the screenshot and also in the UI element list given to you (some elements in the list may NOT be visible on the screen so you can not interact with them).
 - Consider exploring the screen by using the `scroll` action with different directions to reveal additional content.

– The direction parameter for the `scroll` action can be confusing sometimes as it's opposite to swipe, for example, to view content at the bottom, the `scroll` direction should be set to "down". It has been observed that you have difficulties in choosing the correct direction, so if one does not work, try the opposite as well.
Text Related Operations
– Normally to select some text on the screen:  Enter text selection mode by long pressing the area where the text is, then some of the words near the long press point will be selected (highlighted with two pointers indicating the range) and usually a text selection bar will also appear with options like `copy`, `paste`, `select all`, etc. <ii> Select the exact text you need. Usually the text selected from the previous step is NOT the one you want, you need to adjust the range by dragging the two pointers. If you want to select all text in' the text field, simply click the `select all` button in the bar.
– At this point, you don't have the ability to drag something around the screen, so in general you can not select arbitrary text.
– To delete some text: the most efficient way is to directly use the `clear_text` action, which removes all content from a specific text field in one step. Alternatively, the traditional method is to place the cursor at the right position and use the backspace button on the keyboard to delete characters one by one (long pressing backspace can accelerate this if there are many characters). Another approach is to first select the text you want to delete, then press the backspace button on the keyboard.
– To copy some text: first select the exact text you want to copy, which usually also brings up the text selection bar, then click the `copy` button in bar.
– To paste text into a text box, first long press the text box, then usually the text selection bar will appear with a `paste` button in it.
– When typing into a text field, sometimes an auto-complete dropdown list will appear. This usually indicating this is a enum field and you should try to select the best match by clicking the corresponding one in the list.
Now output an action from the above list in the correct JSON format, following the description of the action you are taking. Your answer should look like:
Description: ...
Action: {{"action_type":...}}
Your Answer:

## H.2 PROMPT FOR WORLD MODEL

You are an intelligent autonomous agent designed to predict how a smartphone UI will change in response to specific user actions. You will be given the following:
1. a list of descriptions for most UI elements on the current screen. 2. action1: a user action that to be applied to the current UI. 3. action2: a user action that to be applied to the current UI. action 1 and action 2 are individual actions. Each action is composed of an action type and a corresponding parameter. If the parameter is an index, it refers to the index of a UI element in the provided list. Additionally, assess whether the two actions would result in the same or similar UI changes. Provide a similarity score between 0 and 1 (with two decimal places), where 0 means completely different and 1 means identical.
**Your Task**: For each user action, analyze the current UI and accurately predict the resulting change. Describe each predicted change clearly and in detail.
Here is a list of descriptions for some UI elements on the current screen: ui The user actions are: "action1":pred,"action2":label'
**Output Format**: Return only predicted UI changes as a JSON map of strings as following. Do not include any additional explanations, formatting, or comments.
"action1":"change1...","action2":"change2...","score":"similarity_score"'

### H.3 PROMPT FOR TEACHER MODEL IN THE EXPERIMENTS ON AITW

```
Given a mobile screen and the current context, provide the next action based on the
screen information.
Available Actions (choose one): 1. Dual-point gesture (click): {{"action_type":
"DUAL_POINT", "touch_point": "[y_coord, x_coord]", "lift_point": "[y_coord, x_coord]",
"typed_text": ""}} - For clicking: touch_point and lift_point should be the same
coordinates - Coordinates are normalized between 0.0 and 1.0 (e.g., [0.5, 0.3] means
50% down, 30% right) - Use 4 decimal places for precision (e.g., [0.7761, 0.7089])
2. Dual-point gesture (scroll): {{"action_type": "DUAL_POINT", "touch_point":
"[start_y, start_x]", "lift_point": "[end_y, end_x]", "typed_text": ""}} - For
scrolling up: {{"touch_point": "[0.8, 0.5]", "lift_point": "[0.2, 0.5]"}} - For
scrolling down: {{"touch_point": "[0.2, 0.5]", "lift_point": "[0.8, 0.5]"}} - For
scrolling left: {{"touch_point": "[0.5, 0.8]", "lift_point": "[0.5, 0.2]"}} - For
scrolling right: {{"touch_point": "[0.5, 0.2]", "lift_point": "[0.5, 0.8]"}}
3. Type text: {{"action_type": "TYPE", "touch_point": "[-1.0, -1.0]", "lift_point":
"[-1.0, -1.0]", "typed_text": "<your_text>"}}
4. Navigate back: {{"action_type": "PRESS_BACK", "touch_point": "[-1.0, -1.0]",
"lift_point": "[-1.0, -1.0]", "typed_text": ""}}
5. Navigate home: {{"action_type": "PRESS_HOME", "touch_point": "[-1.0, -1.0]",
"lift_point": "[-1.0, -1.0]", "typed_text": ""}}
6. Press enter: {{"action_type": "PRESS_ENTER", "touch_point": "[-1.0, -1.0]",
"lift_point": "[-1.0, -1.0]", "typed_text": ""}}
7. Task complete: {{"action_type": "STATUS_TASK_COMPLETE", "touch_point": "[-1.0,
-1.0]", "lift_point": "[-1.0, -1.0]", "typed_text": ""}}
Current Context: {processed_obs}
Screen: <See the attached mobile screen image>
Instructions: - Analyze the mobile screen carefully - Choose the most appropriate
action to progress towards the goal - For clicks, estimate the center coordinates
of the target element - For scrolls, choose appropriate direction and distance - For
typing, provide the exact text to be entered - Output only the JSON action format, no
additional text
Answer:
```

## I  QUALITATIVE EXAMPLES

We provide qualitative examples of our SEE modules applied to AitW tasks in Fig. 5 and Fig. 6.

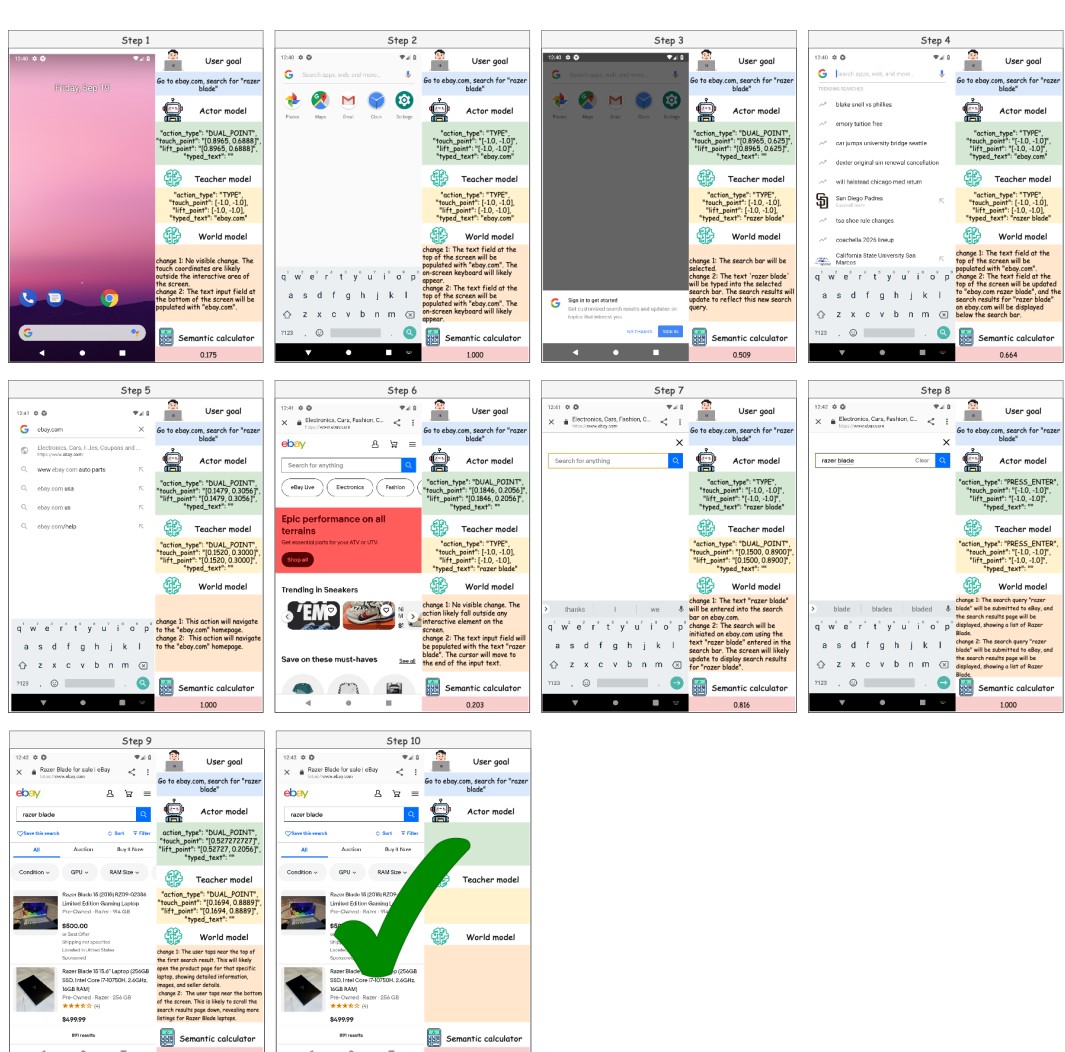

Figure 5: Example of a successful case on an AitW task with our SEE module.

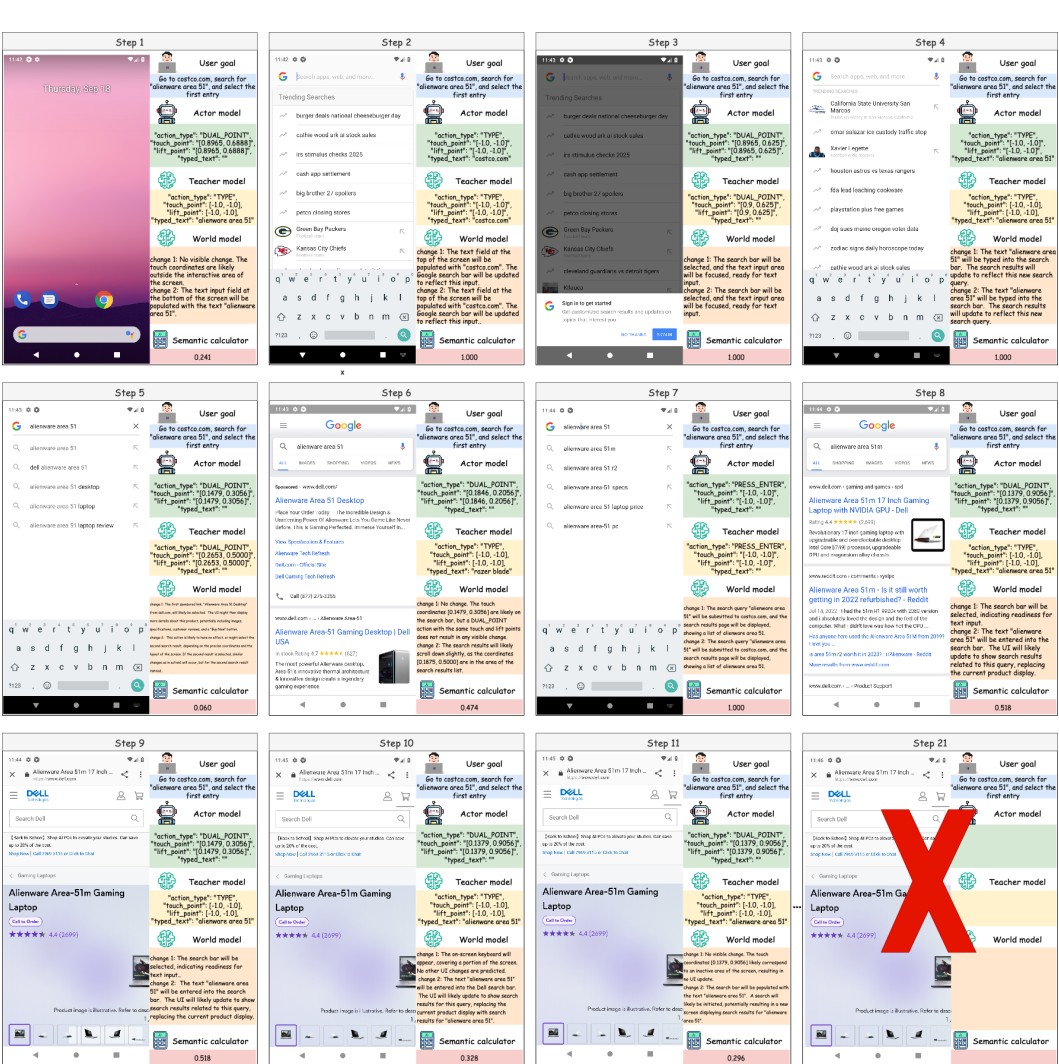

Figure 6: Example of a failure case on an AitW task with our SEE module.

