# OpenReview forum: "BEYOND SYNTAX: ACTION SEMANTICS LEARNING FOR APP AGENTS"
_ICLR.cc/2026/Conference — ICLR 2026 Conference Withdrawn Submission_

### Official Review · Reviewer_hjH7 · 2025-10-21

**Soundness:** 2
**Presentation:** 3
**Contribution:** 2
**Rating:** 4
**Confidence:** 3

**Summary:**

This paper introduces Action Semantics Learning (ASL), a novel learning framework for training smartphone app agents that focuses on understanding the semantic effects of actions rather than their syntactic forms. The framework includes a Semantic Estimator (SEE) that uses a world model to predict UI state changes and BERT to compute semantic similarity between predicted and ground truth actions.

**Strengths:**

The proposed SEE module demonstrates elegant design in being training-free and computationally efficient, avoiding the need for additional GPU memory during deployment while still providing meaningful semantic rewards. The experimental validation is comprehensive, covering both online environments (AndroidWorld, AndroidLab) and offline benchmarks (AndroidControl), with consistent improvements across difficulty levels.

**Weaknesses:**

The reliance on proprietary APIs (Gemini-Flash-2.0) for the world model component undermines the claimed advantages over prompt-based methods that depend on external APIs. This creates a fundamental contradiction: while criticizing existing approaches for API dependency, ASL itself requires API access for its core semantic estimation functionality.

**Questions:**

How does the world model handle dynamic UI elements, time-dependent states, or stochastic UI behaviors that are common in real applications? The paper assumes deterministic state transitions, but many app interactions involve randomness or external factors.

---

### Official Review · Reviewer_nxxF · 2025-10-26

**Soundness:** 2
**Presentation:** 3
**Contribution:** 2
**Rating:** 4
**Confidence:** 4

**Summary:**

This paper focuses on the training of smartphone application agents. It points out that existing methods largely follow a syntax learning paradigm, where models are required to exactly reproduce the *ground-truth action strings* during supervised fine-tuning, which makes them vulnerable to out-of-distribution UI changes.

To address this, the authors propose Action Semantics Learning (ASL) — a new framework that shifts the learning objective from syntactic replication to semantic reproduction of UI state transitions.
Its core component, the Semantic Estimator (SEE), uses an LLM-based world model (Gemini-Flash-2.0) to simulate the UI state changes caused by an action, and then computes the semantic similarity between the predicted and ground-truth transitions using a BERT encoder. The resulting score serves as a reward signal.
ASL can be applied to both supervised fine-tuning and reinforcement learning stages. The authors also provide theoretical analysis suggesting ASL’s robustness under OOD conditions and evaluate it on multiple App-Agent benchmarks.

**Strengths:**

1. The paper correctly identifies that existing App-Agent fine-tuning paradigms overfit to syntactic action forms and generalize poorly to unseen UIs.
2. Drawing inspiration from programming language semantics, the paper reframes action learning as learning the resulting state transitions—a novel and thought-provoking perspective.
3. ASL can be integrated into both SFT and RL pipelines, showing potential for broader applicability.

**Weaknesses:**

The overall effectiveness of ASL hinges on an unverified key assumption: that Gemini-Flash-2.0 (or similar LLMs) can accurately simulate the UI state transition for any given action. However, the paper provides no evaluation of this world model’s prediction accuracy, nor reports its failure rate or bias. If the world model predicts incorrect transitions, SEE will generate misleading semantic rewards—propagating wrong signals to the downstream agent. Such reward noise may reduce robustness rather than improve it. The authors should quantify or analyze how closely the world model’s predicted transitions align with the actual UI state changes; otherwise, the validity of ASL rests on a fragile assumption.

SEE uses BERT cosine similarity between textual descriptions of predicted and ground-truth state transitions. This design is both linguistically sensitive and conceptually imprecise. LLMs may describe the same state change using very different wording, leading to low similarity despite true equivalence. Semantic equivalence is inherently a discrete yes/no relation, but the model treats it as a continuous reward. The paper applies a threshold only in the RL stage but not during SFT, where the continuous score directly modulates the gradient. This could produce non-zero rewards for incorrect actions, potentially destabilizing training. The authors should clarify the normalization and thresholding mechanisms of the reward, otherwise ASL might not be learning true semantic equivalence in practice.

The main experiments compare Qwen-2.5-7B-Instruct-ASL only with its SFT counterpart. This resembles an internal ablation rather than a comparison with state-of-the-art methods. Although Appendix G reports some results against AGUVIS and GUI-R1, this evaluation is narrow—restricted to a small 500-sample subset of AndroidControl. More importantly, for the key online benchmarks (AndroidWorld and AndroidLab), there is no direct comparison with advanced fine-tuning methods. As a result, the empirical evidence remains weak, making it hard to judge whether ASL truly surpasses mainstream approaches under realistic settings.

**Questions:**

1. The validity of ASL depends entirely on Gemini-Flash-2.0’s ability to accurately simulate UI transitions.
  Could you provide quantitative evaluation or qualitative analysis verifying how well these predicted transitions match real ones?
2. How does SEE perform if another LLM (e.g., GPT-4o, Qwen, Claude) is used as the world model? Is the framework robust to such substitutions?
3. If the world model mispredicts transitions, how does this affect the reward signal and convergence of ASL? How is the continuous reward signal handled during SFT? Why is the threshold only introduced in the RL stage but not in SFT?

---

### Official Review · Reviewer_7Fu5 · 2025-10-30

**Soundness:** 2
**Presentation:** 3
**Contribution:** 2
**Rating:** 4
**Confidence:** 4

**Summary:**

This paper introduces something called Action Semantics Learning (ASL) for app agents. The main idea is to focus on what the actions mean in terms of changing the app state, instead of just copying the exact words of the actions. They use a semantic estimator to reward actions that have the same effect, even if they look different. They say it's better for handling new situations and show some experiments.

**Strengths:**

The paper has a good idea about making agents understand the meaning behind actions, not just the syntax. This could help with generalization. The semantic estimator part is clever because it uses existing models like LLM and BERT without needing extra training, which saves time. The experiments cover different setups and models.

**Weaknesses:**

The paper says current methods force agents to copy action strings exactly, but that's not totally true. For example, there's earlier work like AutoDroid (https://arxiv.org/pdf/2308.15272) that already lets models choose from a list of actions instead of generating text, so the criticism is a bit off. Maybe the authors didn't look enough into related work.

The big claim is about handling out-of-distribution cases, but the experiments don't really prove it. The training and test data seem too similar, like from the same apps. There's no real tough OOD test, like with completely new interfaces or big changes, so it's hard to say if the semantic reward really helps in wild scenarios.

Using the semantic reward might be too heavy on computation. Since it relies on LLM predictions and BERT similarities for every training step, it could slow things down a lot, especially for bigger models or datasets. The paper doesn't talk about how long training takes or if it's practical for large-scale use.

The connection to programming language theory feels weak. They mention denotational semantics but don't really use it in a deep way, like for formal proofs or anything. It's more like a fancy comparison without solid backing, which makes the theory part seem vague.

The performance numbers are pretty low compared to other recent methods, like those on the AndroidWorld leaderboard where small models do better. The paper doesn't explain why their results are worse or how to improve them. For instance, success rates around 70% might not be competitive, and there's no discussion on limitations or next steps.

**Questions:**

- Can you discuss the fundamental advantages of ASL over action-selection methods like AutoDroid?
- Can you demonstrate OOD robustness on apps with entirely unseen UI?
- Can you discuss the training time overhead of the semantic reward mechanism?
- Why does ASL underperform compared to recent AndroidWorld leaderboard results?

---

### Official Review · Reviewer_zEB7 · 2025-10-31

**Soundness:** 2
**Presentation:** 2
**Contribution:** 3
**Rating:** 2
**Confidence:** 4

**Summary:**

This work presents Action Semantics Learning (ASL), a framework designed to train models to grasp the semantics of actions rather than simply reproduce the exact ground-truth strings. ASL includes a Semantic Estimator (SEE) module to measure how similar two actions are in terms of the state changes they produce. To compute the semantic reward, ASL adopts Gemini-2.5-Flash to generate textual descriptions of the next GUI state for both the ground-truth and the predicted actions. Then, ASL is evaluated on both online and offline benchmarks.

**Strengths:**

-	The paper identifies a real and often overlooked issue in GUI agent training: different actions can lead to the same next interface state, yet most current methods treat them as distinct due to strict string matching. Framing this as a semantic rather than syntactic learning problem is both intuitive and meaningful.
-	The adoption of world models to simulate state transitions and compute a semantic reward through cosine similarity is neat idea, which combines conceptual clarity with practical feasibility.
-	The authors evaluate across both online and offline benchmarks, and include analyses under SFT and RL settings.

**Weaknesses:**

-	While ASL point out the mismatch between syntax and semantics at the single-action level, it does not extend to cases where multiple action sequences can achieve the same task outcome. This multi-step semantic equivalence is more central to how real GUI agents should reason, and remains outside the current formulation.
-	For both the AndroidLab and WebArena-Lite, task success is judged by another LLM. It inevitably raises questions about the reliability and consistency of the reported results. A small validation study with human raters, or a discussion of the LLM evaluator’s bias, would help strengthen the paper’s credibility.
-	Across most benchmarks, improvements are within a few percentage points, and comparisons are limited to relatively weak baselines. It would be more convincing to include more existing works.

Minor issues:
In line 362, the sentence “The 2K human-generated samples are curated to ensure no overlap with the test set in AndroidControl.” appears inconsistent with the preceding description, which indicates that the 2K samples come from AndroidWorld rather than AndroidControl.

**Questions:**

Please see weaknesses above.

---

### Note · Authors · 2025-11-12

I have read and agree with the venue's withdrawal policy on behalf of myself and my co-authors.